# Energy-Efficient Optimization for Energy-Harvesting-Enabled mmWave-UAV Heterogeneous Networks

**DOI:** 10.3390/e24020300

**Published:** 2022-02-20

**Authors:** Jinxi Zhang, Gang Chuai, Weidong Gao

**Affiliations:** School of Information and Communication Engineering, Beijing University of Posts and Telecommunications, Beijing 100876, China; jinxi@bupt.edu.cn (J.Z.); gaoweidong@bupt.edu.cn (W.G.)

**Keywords:** D2D communication, energy harvesting, mmWave networks, beamwidth selection, power optimization

## Abstract

Energy Harvesting (EH) is a promising paradigm for 5G heterogeneous communication. EH-enabled Device-to-Device (D2D) communication can assist devices in overcoming the disadvantage of limited battery capacity and improving the Energy Efficiency (EE) by performing EH from ambient wireless signals. Although numerous research works have been conducted on EH-based D2D communication scenarios, the feature of EH-based D2D communication underlying Air-to-Ground (A2G) millimeter-Wave (mmWave) networks has not been fully studied. In this paper, we considered a scenario where multiple Unmanned Aerial Vehicles (UAVs) are deployed to provide energy for D2D Users (DUs) and data transmission for Cellular Users (CUs). We aimed to improve the network EE of EH-enabled D2D communications while reducing the time complexity of beam alignment for mmWave-enabled D2D Users (DUs). We considered a scenario where multiple EH-enabled DUs and CUs coexist, sharing the full mmWave frequency band and adopting high-directive beams for transmitting. To improve the network EE, we propose a joint beamwidth selection, power control, and EH time ratio optimization algorithm for DUs based on alternating optimization. We iteratively optimized one of the three variables, fixing the other two. During each iteration, we first used a game-theoretic approach to adjust the beamwidths of DUs to achieve the sub-optimal EE. Then, the problem with regard to power optimization was solved by the Dinkelbach method and Successive Convex Approximation (SCA). Finally, we performed the optimization of the EH time ratio using linear fractional programming to further increase the EE. By performing extensive simulation experiments, we validated the convergence and effectiveness of our algorithm. The results showed that our proposed algorithm outperformed the fixed beamwidth and fixed power strategy and could closely approach the performance of exhaustive search, particle swarm optimization, and the genetic algorithm, but with a much reduced time complexity.

## 1. Introduction

Recently, UAV-assisted communication-network-supporting energy transfer has gained significant attention. With the advantages of flexible deployment and low-cost deployment, UAVs can quickly establish A2G links and transmit information and energy to ground users [1,2]. UAVs have been widely used in emergency communications to quickly restore ground equipment communications in disaster-stricken areas or improve the quality of service for degraded users in overloaded ground areas. UAV Base Stations (UAV-BSs) equipped with large-scale antenna arrays can provide directional transmission of information and energy, thereby avoiding interference between UAVs caused by strong Line-of-Sight (LoS) paths.

As predicted by Cisco, there will be 14.7 billion Machine-to-Machine (M2M) connections and on average 1.8 connections for each member of the global population by 2023 [3]. M2M communication or D2D communication is playing a more important role in the industry of the Internet of Things (IoT), by providing a variety of services, such as smart manufacturing, home automation, video surveillance, etc. In addition, D2D communication can offload traffic for base stations by directly establishing a transmission link from the source to the destination [4]. In mmWave-enabled 5G networks, the high directional beams can improve the link quality and boost the system capacity. However, mmWave networks operating at 30–300 GHz are susceptible to high path loss and are easily blocked by obstacles, which limits the long-distance propagation of signals [5]. D2D devices can act as relays for the blocked mmWave link by establishing multi-hop communication or directly deliver the pre-cached content to the destinations [6,7]. In this case, integrating D2D communication into UAV networks can overcome the problem of the coverage limitation and alleviate the burden of UAVs.

In 5G networks, the transmission based on high directional beams will be widely used, thanks to the wide deployment of mmWave antenna arrays [8]. At the same time, a huge energy consumption caused by large-scale antenna arrays is also incurred, especially for D2D links. Different from ground BSs or UAV-BSs, which have a stable energy supply, D2D devices equipped with massive antenna arrays are prone to suffer energy shortage. In addition, with the rapid increase of data traffic, the energy consumption of D2D devices has become a major concern in the industry, and the Energy Efficiency (EE) of D2D links needs to be urgently optimized. In this case, EH emerges as a promising way to increase the EE of D2D links by providing continuous energy supply for devices and sustaining the stability of links. Equipped with Radio-Frequency (RF) energy harvesters, D2D links are capable of harvesting energy from ambient RF signals while receiving information data. Not only the desired signal, but the interference and noise can be utilized to perform energy harvesting. This extends the battery life for devices and prolongs the communication period for D2D links, which is also beneficial to the improvement of the network EE.

In this paper, we considered an EH-enabled D2D underlying UAV-assisted network scenario, where Simultaneous Wireless Information and Power Transfer (SWIPT) is adopted at the UAVs to enable the transmission of information and energy for ground devices. High directional beams were adopted for Air-to-Ground (A2G) and D2D transmission. While the CUs are receiving the information from the UAV-BSs, D2D transmitters are capable of harvesting energy from the A2G signal. We adopted the EH-BA-DT protocol [9,10] for DUs in EH-enabled scenarios. The system time was divided into the EH phase, Beam Alignment (BA) phase, and Data Transmission (DT) phase. As known to all, the network EE is closely related to the user rate and energy consumption. Therefore, to achieve the maximal EE, a tradeoff between increasing the data rate and decreasing the consumed energy is required. In our scenario, the interference issue was complex, which greatly affects the network EE. Misaligned beams will cause severe interference among users, which will also impact the EE of D2D links. In addition, due to the proliferation of mobile devices and the ever-increasing network traffic, data transmission and signaling exchange quickly consume the energy of D2D links, which makes the limited battery capacity of D2D devices easily depleted. To tackle this problem, an effective method is to alleviate the interference and increase the network EE by performing the joint optimization of beamwidths, transmit power, and EH time ratio for D2D pairs.

The beamwidth of the DU is an important factor affecting the user rate and energy efficiency. For the beam-enabled D2D users, the beam alignment process is first executed to determine the best-matched beam pair at the transmitting and receiving side before the transmission process [11], which means a long beam alignment time will shorten the time for EH and DT, and vice versa. Especially for high-speed mobile users, the process of beam alignment needs to be performed repeatedly. Therefore, controlling the time complexity of beam alignment is helpful to improve the transmission efficiency of the system. Although Exhaustive Search (ES) and Particle Swarm Optimization (PSO) can achieve excellent performance in beam alignment, the time complexity of the algorithms is too large to be effectively applied to the actual network. Therefore, to fully unleash the potential of the mmWave D2D link, it is of great significance to obtain the optimal/sub-optimal beamwidth in a short time to improve the network performance.

Besides, EH emerges as a promising technology to increase the EE of users relying on wireless charging. In traditional low-frequency scenarios, wireless charging has not shown great performance due to the low received signal strength. In 5G mmWave links, multi-antenna system can be employed at the transmitter to achieve effective RF energy harvesting through the enhanced transmission gain brought by beamforming. The authors in [12,13,14] verified the feasibility of the combination of EH and mmWave technology. Although the optimization of the network EE in EH-enabled low-frequency scenarios has been widely studied, there still is a lack of an effective method to improve the network EE in mmWave scenarios. Hence, in this paper, we applied EH to mmWave D2D links and aimed to enhance the battery life of the links by increasing the network EE.

When a D2D link selects a large transmitting and receiving beamwidth, the beam-sweeping process is carried out in a large-angle manner, which will reduce the beam alignment time and increase the time for EH and DT. However, a large transmitting beamwidth will possibly degrade the performance of other links, and a large receiving beam will suffer more serious interference. On the contrary, reducing the transmitting and receiving beamwidth is beneficial to mitigate potential interference, but the time of the BA phase will be greatly prolonged because a huge number of pilot signals are needed to align the beams, which will shorten the time for energy harvesting and data transmission and degrade the system performance.

Moreover, for D2D links, the power and EH time ratio have a significant impact on the network EE. The transmit power not only affects the user’s achievable rate, but also the rate of other D2D links and the overall energy consumption. Therefore, the transmit power for DUs will have a complex impact on the network EE and needs to be optimized. Similarly, the network EE is also closely related to the EH time ratio. Fixing the BA time ratio and increasing the EH time ratio can increase the energy harvested, but this will also reduce the effective transmission time, and vice versa. Moreover, the performance of CUs should be protected by controlling the interference from DUs to CUs.

In this case, to improve the network EE, the three coupling variables, beamwidth, transmit power, and the EH time ratio, need to be considered comprehensively. In this paper, we propose a joint optimization of beamwidth selection, power control, and the EH time ratio to iteratively optimize the variables while fixing the other two. First, a low-complexity beamwidth selection scheme was designed to allow users to find the optimal/sub-optimal beamwidth in a short time. Specifically, DUs first form coalitions to choose different beamwidths. Then, the coalitions are continuously updated along the direction of improving the network utility (i.e., EE) until the final Nash equilibrium is reached. Next, a power control method was designed to suppress the interference and improve the EE. The power levels of D2D transmitters were optimized using the Dinkelbach method and SCA to obtain the sub-optimal power for DUs. Finally, the optimization of the EH time ratio was executed to further improve the EE. The EH time ratio was optimized by solving the convex linear fractional programming. The network EE was iteratively updated until reaching the convergence point.

In conclusion, the main contributions of this paper can be summarized as follows:Different from existing research, we considered a downlink scenario of EH-enabled D2D underlying a UAV-assisted mmWave network and took into consideration the complex interference issue. We built an efficient framework for improving the EE of D2D links by jointly optimizing the beamwidth, transmit power, and EH time ratio of the D2D links using alternating optimization, while guaranteeing the rate requirements of CUs and DUs. The proposed algorithm has low computation complexity and is applicable to large-scale mmWave networks;We constructed a coalition game model to solve the beamwidth selection problem for D2D users, which can explore the potential beamwidth combinations of DUs and converge to the beamwidth structure with sub-optimal utility. A low-complexity beamwidth selection algorithm was proposed to adjust the beamwidths of DUs for increasing the network EE;The transmit power of D2D links was optimized to further improve the system EE. The original non-convex problem with regard to power optimization was solved by non-fractional programming and successive convex approximation. Specifically, we first transformed the non-convex fractional programming into a non-fractional problem using the Dinkelbach method. Next, we eliminated the non-convexity in the formula by approximating the non-convex part with its first-order Taylor expansion. Along with the power, we also optimized the EH time ratio. The problem with regard to the EH time ratio was proven to be a convex linear fractional programming, and hence, the optimal solution is readily obtained;We validated the convergence and effectiveness of the proposed algorithm by performing numerical experiments with different settings for the network parameters. The proposed algorithm can converge to the optimal EE after finite iterations and effectively improve the network EE compared with the Fixed Beamwidth and Fixed Power (FBFP) scheme. Our proposed algorithm can achieve performance close to ES and PSO, but with much reduced complexity. The factors that potentially affect the network EE, such as the number of DUs, the number of CUs, the maximal power for DUs, the minimum harvested energy, and the minimum rate for CUs, are also thoroughly discussed and analyzed.

The rest of this paper is organized as follows: The related works are given in Section 2. In Section 3, the EH-enabled D2D underlying the UAV network model and problem formulation are presented in detail. In Section 4, the alternating optimization method is employed to jointly optimize the beamwidth, transmit power, and EH time ratio to obtain the sub-optimal EE with a low time complexity. The simulation results are shown and discussed thoroughly in Section 5. Finally, the conclusions are drawn in Section 6.

## 2. Related Works

Under the mmWave network architecture, energy harvesting is capable of providing reliable and sustainable energy coverage and shows better performance than conventional low-frequency scenarios [12,13,14]. The authors of [15,16] investigated the performance of energy harvesting in UAV-assisted mmWave technology. The authors of [15] derived the energy and SINR coverage probability under the UAV mmWave network enabled by energy harvesting. In [16], the authors analyzed the energy and SINR coverage probability in a hybrid network where sub-6G transmission and mmWave transmission coexist. In [17], the trajectories of two UAVs for data gathering and energy transferring were optimized using the deep reinforcement learning approach to reduce the system energy consumption and improve the timeliness of service for users. The trajectory optimization was performed in [18] to minimize the energy consumption of the UAV and guarantee the user rate. In [19], the authors considered the effect of beam alignment error in an EH-enabled mmWave network and derived the energy coverage probability by assuming a non-linear EH process.

Moreover, energy harvesting can provide new possibilities for improving the performance of D2D communication [20], which has been the focus of academia [21,22,23,24,25,26]. In [21], the authors considered a non-linear energy harvesting model and classified the users into the EH group and the non-EH group based on the minimum harvested power threshold. Then, the resource allocation and power control were iteratively optimized to maximize the sum-EE using non-fractional programming and the Lagrange method. The authors of [22] proposed a power control and time scheduling algorithm to improve the capacity of the system under a time splitting architecture, where each BS was equipped with a single antenna, and they further reformed the algorithm and applied it to the scenario where the BSs were equipped with multiple antennas in [23]. The authors in [24] used a stochastic model to derive the ergodic capacity of EH-enabled D2D communication and proposed an effective mode selection method to improve the system EE. The outage issue of EH-based D2D communication has also been widely studied [25,26]. In [25], the outage probability of D2D links was analyzed considering the spectrum sharing between DUs and CUs. A power control scheme for DUs while ensuring the outage probability for CUs below the threshold was proposed in [26].

In mmWave networks, another issue that attracts the attention of academia is the optimization of the beamwidth. To achieve the tradeoff between alleviating interference and reducing beam alignment complexity, there have been many research works on beamwidth selection in mmWave networks [9,10,11,27,28,29,30,31,32,33]. In [11], we proposed an coalition-game-based beamwidth selection algorithm for mmWave-enabled D2D links. In [27], the authors proposed two joint beamwidth selection and scheduling schemes. The authors exploited the method of interference estimation rather than precise calculation to measure the beam-level interference among the users, which achieved much improved throughput compared with existing standards [34]. In [28], the authors revealed the relationship between latency and overhead in the beam alignment process, which indicated that multi-beam simultaneous scan can provide the best tradeoff between latency and overhead. A recent research work [29] used the geometry model to characterize the beamwidth selection and obtained the near-optimal solution for the beamwidths of users in mmWave networks. In [30], the authors considered a simple full-duplex mmWave wireless network architecture and exploited the numerical solution to obtain the optimal beamwidth for the users to maximize the network energy efficiency. In D2D communication, the beamwidth design for devices is also a research hotspot. In [9,10], the authors considered a single D2D pair in a mmWave network without considering the inter-user interference. The throughput and EE were maximized in [9,10], respectively, by optimizing the beamwidth and EH ratio of the D2D pair. To overcome the complexity of ES, PSO has also been applied in beamwidth optimization of V2V communication [31] and D2D communication [32,33] to obtain the sub-optimal solution.

## 3. System Model and Problem Formulation

### 3.1. Network Topology

As shown in Figure 1, we considered an EH-enabled D2D underlying the mmWave network consisting of multiple mmWave UAV-BSs (UBSs). *M* cellular users and *N* D2D users were randomly distributed in the considered area. Each D2D link was composed of a D2D Transmitter (DT) and D2D Receiver (DR). There exist two kinds of communication links in the network: A2G links and D2D links. A2G links denote the transmission from the UAV to ground users, including cellular users and the D2D transmitter. D2D links denote the direct transmission from the DT to the DR. It was assumed that both the A2G links and D2D links adopt narrow beams for transmission. We denote the set of CUs and DUs as $C=\{C_{1},C_{2},\ldots ,C_{M}\}$ and $D=\{D_{1},D_{2},\ldots ,D_{N}\}$, respectively. For the *n*th D2D link, we denote its DT and DR as $D_{n}^{T}$ and $D_{n}^{R}$. In our scenario, both A2G links and D2D links reuse the full mmWave frequency band and adopt narrow beams for transmission. The UAVs are connected to the ground gateway to transmit the backhauling data. In addition, we assumed that data collection and algorithm execution were performed at the Operation And Management (OAM) in the gateway. We assumed that the coverage of the UAV is determined based on the path loss threshold [35], and the association between UAVs and users was established based on minimum path loss criterion. In addition, we ignored the movement of UAVs and users and assumed all the UAVs hovered at a fixed horizontal position and fixed altitude. However, our algorithm can also be utilized in a dynamic and fast-changing environment since the time frame can be approximately divided into multiple snapshots, in which our system model and algorithm can be applied.

In our paper, we did not focus on the energy consumption of the UAVs, since they have a larger battery capacity and can be charged by a charging station. In our scenario, we incorporated EH technology into the system and assumed that each DT can harvest energy from the ambient RF signals to prolong the transmission time. We assumed that the A2G links, including signal and interfering signals, are utilized by the DTs to harvest energy and restore the energy of the battery. As shown in Figure 2, we adopted the EH-BA-DT protocol for D2D links, where the total transmission time for $D_{n}$ with duration *T* is divided into three phases: (1) energy harvest phase with a duration of $\tau_{e}T$: DTs harvest the energy from the ambient environment and store it in the battery; (2) beam alignment phase with a duration of $\tau_{n}^{ba}T$, during which the DUs perform beamwidth alignment for the transmitter and receiver; (3) data transmission phase with a duration of $(1-\tau_{e}-\tau_{n}^{ba})T$: after EH and BA, the DT starts transmitting data to its associated DR, where $\tau_{e}$ is the unified time ratio of EH for all the DUs, and $\tau_{n}^{ba}$ is the time ratio of BA for $D_{n}$, respectively.

### 3.2. Alignment Delay

In our model, we only considered the optimization of beamwidths for D2D links, assuming that the beamwidths of A2G links have been determined and vary with time. According to [7], to avoid the high time complexity of narrow beam search in the whole angle range, a sector-level alignment with a large angle sweeping was first performed. Then, the beam-level alignment with a much more refined beam search was conducted in the aligned sector, which obtained the best matched beam pair at the transmitting and receiving side. Hence, the beam alignment time ratio for D2D link $D_{n}^{T}\to D_{n}^{R}$ is:(1)$$\tau_{n}^{ba}=\left(\theta_{n,n}^{t}\theta_{n,n}^{r}/\phi_{n,n}^{t}\phi_{n,n}^{r}\right)T_{p}/T$$
where $\theta_{n,n}^{t}$ and $\theta_{n,n}^{r}$ denote the sector-level beamwidth at $D_{n}^{T}$ and $D_{n}^{R}$ and $T_{p}$ denotes the pilot transmission time. For analytical tractability, we assumed the beamwidths for $D_{n}^{T}$ and $D_{n}^{R}$ were equal: $\phi_{n}=\phi_{n,n}^{t}=\phi_{n,n}^{r}$, which is beneficial to speed up the beam alignment process. In addition, we assumed that $\theta =\theta_{n,n}^{t}=\theta_{n,n}^{r},\forall n\in N$, which means the sector-level beamwidths for all the DUs are also equal. Let BW denote the feasible beamwidth set, which contains all the feasible beamwidths that each D2D pair can select, then the range of BW is denoted as follows:(2)$$max\left(ceil\left(\sqrt{\theta^{2}T_{p}/T}\right),\phi_{min}\right)\leq {\mathrm{BW}}_{i}\leq \theta ,\;\forall n\in N$$
where ceil(x) denotes the ceiling function, ${\mathrm{BW}}_{i}$ is an element in the feasible beamwidth set BW, and $\phi_{min}$ is the minimum beamwidth for D2D pairs.

### 3.3. Channel Modeling

We assumed that both D2D links and A2G links had full access to the whole mmWave bandwidth and the interference can be avoided to a great extent thanks to high directional beams. However, once the beams of interferers and receivers are accidentally aligned, severe interference is introduced. Moreover, there also exists sidelobe interference in the network, which is trivial, but not negligible.

We adopted the sectored antenna model presented in [36] to calculate the antenna gain of mmWave links, where the antenna gain of the beam-steered transmitter and receivers is constant for all the angles in the mainlobe, and the antenna gain in the sidelobe was also regarded as a small constant 0<z≪1. Let $\varphi_{a,b}^{t}$ and $\varphi_{a,b}^{r}$ denote the alignment error angle at the transmitter and receiver, and the transmitting beam gain at transmitter *a* is calculated as:(3)$$g_{a,b}^{t}=\left\{\begin{array}{cc} \frac{2\pi -(2\pi -\phi_{a,b}^{t})z}{\phi_{a,b}^{t}}, & if\;\varphi_{a,b}^{t}\leq \phi_{a,b}^{t}.\\ z, & otherwise. \end{array}\right.$$
where $\phi_{a,b}^{t}$ is the mainlobe beamwidth selected by transmitter *a*. Similarly, the receiving beam gain at receiver *b* is calculated as (Equation 4), where $\phi_{a,b}^{r}$ is the receiving beamwidth at receiver *b*:(4)$$g_{a,b}^{r}=\left\{\begin{array}{cc} \frac{2\pi -(2\pi -\phi_{a,b}^{r})z}{\phi_{a,b}^{r}}, & if\;\varphi_{a,b}^{r}\leq \phi_{a,b}^{r}.\\ z, & otherwise. \end{array}\right.$$

For A2G links, the channel gain between the UAV *m* and receiver *k* (CU or DT) is calculated as $g_{U_{m},k}^{c}=10^{-PL_{U_{m},k}/10}$, where $PL_{U_{m},k}$ is the path loss between UAV $U_{m}$ and user *k*, which follow the free-space path loss model due to the high probability of the LoS path:(5)$$PL\left(d_{U_{m},k}\right)\left(dB\right)=d_{0}/d_{U_{m},k}^{2}$$
where $d_{0}$ is the channel gain of reference distance 1m and $d_{U_{m},m}$ is the distance between user *m* and UBS $U_{m}$. Hence, the total channel gain between $U_{m}$ and *k* is:(6)$$h_{U_{m},k}=g_{U_{m},k}^{t}g_{U_{m},k}^{c}g_{U_{m},k}^{r}$$

For ground links, the channel gain of link *a* (DT) →*b* (DR or CU) is calculated as $g_{a,b}^{c}=10^{-PL_{a,b}/10}$, where $PL_{a,b}$ denotes the path loss of link a→b, which is modeled as [36]:(7)$$PL_{a,b}\left(dB\right)=20\log_{10}\left(\frac{4\pi d_{0}}{\lambda }\right)+10A\mathrm{lg}\left(d_{a,b}/d_{0}\right)+\chi_{SF}$$
where $d_{a,b}$ is the horizontal distance from the transmitter *a* to the receiver *b* of each link. λ and *A* denote the wavelength and the path loss exponent. $\chi_{SF}$ is the shadow fading factor, which follows a Gaussian distribution $\chi_{SF}∼N\left(0,\sigma_{SF}^{2}\right)$.

Hence, the total link gain from a→b is calculated as the product of the transmitting gain, channel gain, and receiving gain:(8)$$h_{a,b}=g_{a,b}^{t}g_{a,b}^{c}g_{a,b}^{r}$$

For cellular user $C_{m}$, the interference comes from the other UBSs and D2D transmitters in the network, and its Signal-to-Interference-plus-Noise Ratio (SINR) is calculated as:(9)$$SINR_{m}^{C}=\frac{P_{UBS_{m}}h_{BS_{m},m}}{I_{C}^{m}+I_{D}^{m}+N_{0}}$$
where $P_{UBS_{m}}$ is the transmit power of $UBS_{m}$, which is the serving UBS of $C_{m}$. $I_{C}^{m}=\sum_{i\in M\setminus U_{m}}\sum_{j\in C_{i}}P_{UBS_{i}}h_{UBS_{i,j},m}$ denotes the interference from other UBSs in the network, where $C_{i}$ is the set of serving CUs of UAV *i*. $h_{UBS_{i,j},m}$ is the channel gain between the transmit beam of UAV *i* for the *j*-th user and the receive beam of D2D receiver *m*. $I_{D}^{m}=\sum_{i\in N}p_{i}h_{D_{i}^{t},m}$ denotes the interference from D2D transmitters, where $p_{i}$ is the transmit power of $D_{i}^{T}$. $N_{0}$ is the noise power.

For D2D pair $D_{n}$, its SINR can be calculated as:(10)$$SINR_{n}^{D}=\frac{p_{n}h_{D_{n}^{T},D_{n}^{R}}}{I_{C}^{n}+I_{D}^{n}+N_{0}}$$
where $I_{C}^{n}=\sum_{i\in M}\sum_{j\in C_{i}}P_{UBS_{i}}h_{UBS_{i,j},D_{n}^{r}}$ and $I_{D}^{n}=\sum_{i\in N\setminus n}p_{i}h_{i,n}$ denote the interference from cellular links and other D2D transmitters, respectively.

For EH-enabled DTs, assuming the energy harvesting efficiency is γ, the harvested energy at $D_{n}^{T}$ is calculated as:(11)$$EH_{n}=\gamma \tau_{e}T\left(\sum_{i\in M}\sum_{j\in C_{i}}P_{UBS_{i}}h_{UBS_{i,j},D_{n}^{t}}+N_{0}\right)$$

Then, the total consumed power during the transmission time *T* for $D_{n}$ is:(12)$$P_{n}^{con}=2P_{cir}T+p_{n}\left(1-\tau_{n}^{ba}-\tau_{e}\right)T-EH_{n}$$
where $P_{cir}$ is the circuit power consumption at $D_{n}^{t}$ and $D_{n}^{r}$. For simplicity, we omitted the power consumption for DUs in the BA phase as it is trivial when compared to the amount of harvested energy and consumed energy for transmitting data.

According to the Shannon formula, the achievable rate of CUs and DUs is expressed as (Equation 13) and (Equation 14), respectively:(13)$$R_{m}^{C}=\log_{2}\left(1+SINR_{m}^{C}\right)$$
(14)$$R_{n}^{D}=\left(1-\tau_{e}-\tau_{n}^{ba}\right)\log_{2}\left(1+SINR_{n}^{D}\right)$$

Under these assumptions, we can formulate our target problem. Without loss of generality, we express the network EE as the ratio of the sum of the user rate and the sum of the energy consumed by DUs. Let $\phi =\{\phi_{n},\forall n\in N\}$ and $p=\{p_{n},\forall n\in N\}$ denote the beamwidth vector and power vector of D2D links, then the problem of maximizing the EE of D2D links by optimizing ϕ, p, and $\tau_{e}$ is formulated as:(15)$$\begin{array}{cc} \underset{\phi ,p,\tau_{e}}{maximize} & \;\eta =\frac{\sum_{n\in N}R_{n}^{D}}{\sum_{n\in N}P_{n}^{con}}\\ s.t.\;\;C1 & :R_{n}^{D}\geq R_{min}^{D},\forall n\in N\\ C2 & :R_{m}^{C}\geq R_{min}^{C},\forall m\in M\\ C3 & :max\left(ceil\left(\sqrt{\theta^{2}T_{p}/T}\right),\phi_{min}\right)\leq \phi_{n}\leq \theta ,\;\forall n\in N\\ C4 & :0\leq p_{n}\leq p_{D}^{max},\forall n\in N\\ C5 & :\tau_{n}^{ba}+\tau_{e}\leq 1,\forall n\in N\\ C6 & :\tau_{e}\geq 0\\ C7 & :EH_{n}\geq EH_{min},\forall n\in N \end{array}$$
where C1 and C2 guarantee the minimum rate requirements for DUs and CUs, respectively. C3 indicates the feasible range for the transmitting and receiving beamwidths of the D2D links. C4 is the power constraint for the DUs. C5 and C6 ensure that the time ratios for beam alignment, energy harvesting, and data transmission are all positive. C7 indicates that the harvested energy for each DU should exceed the minimum amount to activate the EH process.

## 4. Proposed Algorithm

The problem (Equation 15) is difficult to solve and computationally hard, especially when a huge number of D2D links and CUs exist in the network. In this section, we resorted to the alternating optimization, also known as the block coordinate descent method, to alternately optimize one of the variables given the other two. To be specific, we first optimized the beamwidths ϕ, fixing p and $\tau_{e}$. Next, we applied the Dinkelbach method and successive convex approximation to obtain the sub-optimal solution of the transmit power p, fixing the beamwidth ϕ and time ratio $\tau_{e}$. Finally, we optimized $\tau_{e}$ by solving the standard linear fractional programming, given the beamwidth ϕ and transmit power p.

### 4.1. Coalition-Game-Based Beamwidth Selection Algorithm

Given transmit power p and EH time ratio $\tau_{e}$, the problem (Equation 15) can be reduced to the following problem:(16)$$\begin{array}{cc} \underset{\phi }{maximize} & \;\eta =\frac{\sum_{n\in N}R_{n}^{D}}{\sum_{n\in N}P_{n}^{con}}\\ \; & s.t.\;\;C1,C2,C3,C5\;and\;C7\; \end{array}$$

However, due to the huge number of feasible beamwidths for DUs, Problem (Equation 16) is still hard to solve. Although Exhaustive Search (ES) can solve the problem optimally, its time complexity is unbearable; thus, its practicability is limited. When the number of DU grows, the time complexity of ES grows exponentially, which incurs unacceptable overhead. Hence, we resorted to a coalition game to obtain the sub-optimal solution of Problem (Equation 16), which is time efficient and can achieve performance close to ES.

To solve the original problem, we formulated a coalition game G={P,X,U}, where P=N denotes the player set formed by DUs, X is the strategy space, which contains all the strategies that players can adopt, and *U* is the transferable utility. In the proposed game, multiple players (i.e., D2D pairs) forming a coalition can be regarded as choosing the corresponding beamwidth; thus, a coalition structure is established, which also corresponds to a beamwidth strategy. The number of coalitions is the same as the number of feasible beamwidths, i.e., $\left|\mathrm{BW}\right|$, and each coalition consists of a group of players that select the same beamwidth. The ultimate goal of the coalition game is to find the Nash-stable coalition structure with the optimal/sub-optimal system utility.

Let $F=\{F_{1},F_{2},\ldots ,F_{\left|\mathrm{BW}\right|}\}$ denote the coalition structure that all the players form and $F_{c}\in F$ be a coalition that a group of players forms, which indicates these DUs select the same beamwidth. The coalitions are non-overlapping, and the entire coalition structure should contain all the DUs, which means that $F_{i}\cap F_{j}=⌀$, for any i≠j, and $\underset{i\in \{1,2,\ldots ,\left|\mathrm{BW}\right|\}}{\cup }F_{i}=N$.

Aiming at improving the EE while guaranteeing the performance of the CU, the utility under structure F is calculated as the overall EE of the D2D links:(17)$$U\left(F\right)=\left\{\begin{array}{cc} \eta , & if\;C1,C2,C3,C5\;and\;C7\;are\;met.\\ -inf, & else. \end{array}\right.$$It can be seen from (Equation 17) that our ultimate goal was to maximize the overall EE of the D2D links rather than focusing on the individual utility. In addition, any beamwidth structure that fails to satisfy the constraints in (Equation 16) will be given a utility of negative infinity as a penalty.

**Definition** **1**(preference order ). *For player $D_{n}\in N$, the preference order ${▹}_{n}$ is defined as a complete, reflexive, and transitive binary relation over all the coalitions that $D_{n}$ can possibly join. For player $D_{n}\in N$, given two coalition structures F and $F^{\prime }$ and two coalitions $F_{c}\in F$ and $F_{c^{\prime }}\in F^{\prime }$, $F_{c}{▹}_{n}F_{c^{\prime }}$ indicates that $D_{n}$ prefers being a member of $F_{c}$ to form structure F than being a member of another coalition $F_{c^{\prime }}$ to form structure $F^{\prime }$, i.e., $D_{n}$ prefers selecting beamwidth $F_{c}$ rather than $F_{c^{\prime }}$ to improve the network EE. The switch rule that determines the preference order for the players is defined as:*
(18)$$F_{c}{▹}_{n}F_{c^{\prime }}\Leftrightarrow U\left(F\right)>U\left(F^{\prime }\right)$$
*This switch rule (Equation 18) demonstrates that: to improve the network EE, player $D_{n}$ prefers being a member of coalition $F_{c}$ than being a member of $F_{c^{\prime }}$.*

**Definition** **2**(switch operation ). *Given a coalition structure $F=\{F_{1},F_{2},\ldots ,F_{c}\ldots ,F_{c^{\prime }}\ldots ,F_{\left|\mathrm{BW}\right|}\}$, if player $D_{n}$ chooses to leave its current coalition $F_{c}$ and switch to another coalition $F_{c^{\prime }}$, the coalition structure will be updated: $F^{\prime }=\left\{F\setminus \left(F_{c},F_{c^{\prime }}\right)\cup \left(F_{c}-D_{n}\right)\cup \left(F_{c^{\prime }}+D_{n}\right)\right\}$.*

At the initialization stage of the alternating optimization, the coalition structure is initialized satisfying the constraints in Problem (Equation 16). In the following iterations, the coalition structure is first set according to the optimized beamwidth solution of the last iteration. Then, the players are randomly chosen to perform the switch operation. If the switch rule (Equation 18) is strictly satisfied, the selected player will leave $F_{c^{\prime }}$ and join $F_{c}$ to form a new structure. After continuous switch operations performed by players, the coalition structure will be updated and finally converge to the Nash-stable structure, which implies that there is no player who has the incentive to change its beamwidth (i.e., coalition) and form a new coalition structure; thereby, the system utility can no longer be improved. The detailed illustration of Coalition-Game (CG)-based beamwidth selection algorithm is shown in Algorithm 1.

The proposed coalition game can converge after a finite number of iterations. The reason lies in that the number of feasible beamwidths that players can select and the number of players (D2D pairs) are both finite. Therefore, the cardinality of the beamwidth strategy space is also finite. In other words, the number of structures that players can form is limited. Since each switch operation performed by players will possibly visit a new coalition structure, we can reach the conclusion that the switch process will terminate and the final coalition structure will be ultimately reached. Next, we prove the stability of our proposed CG-based beamwidth selection algorithm.
**Algorithm 1** Coalition-game-based beamwidth selection algorithm for D2D pairs during each iteration1:**Initialize**$\left(\phi ,p,\tau_{e}\right)$ with the output of the last iteration, and initialize $F_{ini}=\left\{F_{1},F_{2},\ldots \right\}$ as indicated by ϕ.2:Set current coalition structure $F_{cur}=F_{ini}$.**Require:**3:Randomly select a player $D_{n}\in N$, and denote its current coalition as $F_{c}$.4:$D_{n}$ randomly chooses another coalition $F_{c^{\prime }}\in F_{cur}$, and denote the temporary structure after $D_{n}$ leaves $F_{c}$ and switches to $F_{c^{\prime }}$ as:5:$F_{temp}=F_{cur}\setminus \left(F_{c},F_{c^{\prime }}\right)\cup \left(F_{c}-D_{n}\right)\cup \left(F_{c^{\prime }}+D_{n}\right)$.6:Calculate $U(F_{cur})$ and $U(F_{temp})$.7:**if**$U\left(F_{temp}\right)>U\left(F_{cur}\right)$**then**8:    $D_{n}$ leaves $F_{c}$ and joins $F_{c^{\prime }}$.9:    Update the coalition structure as: $F_{cur}=F_{temp}$.10:**else**11:    $D_{n}$ remains in its current coalition $F_{c}$.**Ensure:** The final Nash-stable coalition structure is reached.
12:**Output** The optimal beamwidths ϕ and $\left(p,\tau_{e}\right)$ to the next block.


**Theorem** **1.**
*The final coalition structure $F^{fin}$ in the proposed algorithm is a Nash-stable coalition structure.*


**Proof.** The final coalition structure is a Nash-stable structure if the system utility can no longer be improved by any player changing its beamwidth, i.e., $F_{c}^{fin}=\underset{F_{c}}{argmax}U\left(F\right),$$\forall F_{c}\in F$, where $F^{fin}=\left\{F_{1}^{fin},F_{2}^{fin},\ldots ,F_{c}^{fin},\ldots ,F_{\left|\mathrm{BW}\right|}^{fin}\right\}$ is the final coalition structure. To prove the Nash-stability of $F^{fin}$, we resorted to a contradiction: if the final formed structure $F^{fin}$ is not stable, which is equivalent to that there is at least a player $D_{n}\in D$, who is in coalition $F_{c}^{fin}$, who will switch to another coalition $F_{c^{\prime }}^{fin}$, due to $\left(F_{c^{\prime }}^{fin}\cup D_{n}\right){▹}_{n}F_{c}^{fin}$. In this case, a new coalition structure is formed, which is contrary to our assumption that $F^{fin}$ is the final coalition structure. So far, the proof that $F^{fin}$ is Nash-stable is complete.    □

### 4.2. Power Optimization for D2D Links

Given beamwidth ϕ and EH time ratio $\tau_{e}$, (Equation 15) can be written as the problem with regard to power optimization:(19)$$\begin{array}{cc} \underset{p}{maximize} & \;\eta =\frac{\sum_{n\in N}R_{n}^{D}}{\sum_{n\in N}P_{n}^{con}}\\ s.t. & \;\;C1,C2,C4\;and\;C7\; \end{array}$$

It can be seen that Problem (Equation 19) is a fractional programming, which is non-convex and hard to solve. Thus, we applied the Dinkelbach method [37] to transform Problem (Equation 19) into a non-fractional problem. Given η, let $R^{D}=\sum_{n\in N}R_{n}$ and $P^{con,D}=\sum_{n\in N}P_{n}^{con}$, then the original problem (Equation 19) can be equivalently transformed into the following problem:(20)$$\begin{array}{c} \underset{p}{maximize}\;R^{D}-\eta P^{con,D}\\ s.t.\;\;C1,C2,C4\;and\;C7\; \end{array}$$

**Theorem** **2**([37]). *The optimal $\eta^{*}$ can be obtained if and only if $R^{D*}-\eta^{*}P^{con,D*}=0$ where:*
(21)$$\begin{array}{c} \eta^{*}=\underset{p}{max}\;\eta =\frac{R^{D}}{P^{con,D}} \end{array}$$
*$R^{D*}$ and $P^{con,D*}$ are the optimal value of $R^{D}$ and $P^{con,D}$, respectively, when η is maximized.*

The proof of Theorem 2 can be referred to [37], and we omit it here. It can be seen from Theorem 2 that we can obtain the optimal solution of Problem (Equation 19) by solving the equivalent Problem (Equation 20).

However, Problem (Equation 20) is still not convex due to the existence of $R^{D}=\sum_{n\in N}R_{n}^{D}$. Next, to eliminate the non-convexity of Problem (Equation 20), we first decomposed $R_{n}^{D}$ in the objective function into the subtractive form:(22)$$\begin{array}{cc} R_{n}^{D}= & \underset{R_{n}^{D^{\prime }}}{\underset{︸}{c_{n}\log_{2}\left(p_{n}g_{n,n}+\sum_{j\in N,j\neq n}p_{j}h_{j,n}+I_{C}^{n}+N_{0}\right)}}\\ - & \underset{R_{n}^{D^{\prime \prime }}}{\underset{︸}{c_{n}\log_{2}\left(\sum_{j\in N,j\neq n}p_{j}h_{j,n}+I_{C}^{n}+N_{0}\right)}} \end{array}$$
where $c_{n}=1-\tau_{n}^{ba}-\tau_{n}^{e}$ denotes the time ratio for data transmission. It can be seen that $R_{n}^{D^{\prime }}$ and $R_{n}^{D^{\prime \prime }}$ are both concave over p; however, the subtraction of two concave functions is not convexity preserving [38]. Hence, given a local point $p^{l}=\left(p_{1}^{l},p_{2}^{l},\cdots ,p_{N}^{l}\right)$, we can exploit the property that a concave function is upper-bounded by its first-order Taylor Expansion. For each n∈N:(23)$$\begin{array}{cc} R_{n}^{D^{\prime \prime }} & \leq R_{n,ub}^{D^{\prime \prime }}\\ \; & =c_{n}(\sum_{j\in N,j\neq n}\frac{\log_{2}^{e}h_{j,n}\left(p_{j}-p_{j}^{l}\right)}{\sum_{m\in N,m\neq n}p_{m}^{l}h_{m,n}+I_{C}^{n}+N_{0}}\\ \; & +\log_{2}\left(\sum_{j\in N,j\neq n}p_{j}^{l}h_{j,n}+I_{C}^{n}+N_{0}\right)) \end{array}$$

On this basis, we can approximate $R_{n}^{D}$ as follows:(24)$$\begin{array}{cc} R_{n}^{D}\geq & R_{n}^{D^{\prime }}-R_{n,ub}^{D^{\prime \prime }}\\ = & R_{n}^{D^{\prime }}-c_{n}\sum_{j\in N,j\neq n}\frac{\log_{2}^{e}h_{j,n}\left(p_{j}-p_{j}^{l}\right)}{\sum_{j\in N,j\neq n}p_{j}^{l}h_{j,n}+I_{C}^{n}+N_{0}}\\ - & c_{n}\log_{2}\left(\sum_{j\in N,j\neq n}p_{j}^{l}h_{j,n}+I_{C}^{n}+N_{0}\right)\\ ≜ & R_{n,lb}^{D} \end{array}$$

The non-convexity in $R_{n}^{D}$ was eliminated by approximating it as $R_{n,lb}^{D}$, and $R^{D}$ in the objective function of (Equation 20) can also be approximated by $R_{lb}^{D}=\sum_{n\in N}R_{n,lb}^{D}$, which provides a lower bound of $R^{D}$.

After approximating each $R_{n}^{D}$ in $R^{D}$, C1 can be transformed into the following concave form:(25)$$\begin{array}{c} C1:R_{n,lb}^{D}\geq R_{min}^{D},\forall n\in N \end{array}$$

Further, the constraint C2 in (Equation 15) can be rewritten as the following form:(26)$$\begin{array}{cc} C2:\sum_{i\in N}p_{i}h_{i,m}+I_{C}^{m}+N_{0}\leq (P_{UBS_{m}}h_{UBS_{m},m}/( & 2^{R_{min}^{C}}-1)),\forall m\in M \end{array}$$

Evidently, C2 is concave over p, and the non-convexity of (Equation 20) has been eliminated so far. To obtain the sub-optimal solution, we can iteratively optimize $R_{lb}^{D}$ by solving the following approximated problem:(27)$$\begin{array}{c} \underset{p}{max}\;R_{lb}^{D}-\eta P^{con,D}\\ s.t.\;\;C1,C2,C4\;and\;C7\; \end{array}$$

Problem (Equation 27) is a convex optimization of p and can be readily solved by the interior point method [39]. The detailed procedure for optimizing p is summarized in Algorithm 2.
**Algorithm 2** SCA-based power optimization for D2D links during each iteration1:**Initialize**$\left(\phi ,p,\tau_{e}\right)$ with the output of Algorithm 1. Set the current iteration l=0 and the initial local point $p^{l}=p$.2:**while** The improvement of the objective function in (Equation 27) is higher than a predefined threshold $\epsilon_{1}$
**do**.3:    Solve Problem (Equation 27) for a given η using the interior point method and obtain the optimal solution $p^{l*}$.4:    Update $\eta =R_{lb}^{D}\left(p^{l*}\right)/P^{con,D}\left(p^{l*}\right)$.5:    Set $p^{l+1}=p^{l*}$ and update l=l+1.6:**Output** the optimal power strategy $p^{l*}$ and $\left(\phi ,\tau_{e}\right)$ to the next block.

### 4.3. Time Scheduling Optimization

Given ϕ and p, the optimization of $\tau_{e}$ is formulated as:(28)$$\begin{array}{cc} \; & \underset{\tau_{e}}{max}\;\eta \\ s.t.\; & C1,C5,C6\;and\;C7\; \end{array}$$

Problem (Equation 28) is a standard linear fractional programming and can be written in the following form:(29)$$\begin{array}{cc} \; & \underset{\tau_{e}}{max}\frac{A\tau_{e}+B}{C\tau_{e}+D}\\ s.t.C1 & :\tau_{e}\leq \min_{n\in N}\left(1-\tau_{n}^{ba}-R_{D}^{min}/\log_{2}\left(1+SINR_{n}^{D}\right)\right)\\ C5 & :\tau_{e}\leq \min_{n\in N}\left(1-\tau_{n}^{ba}\right)\\ C6 & :\tau_{e}\geq 0\\ C7 & :\tau_{e}\geq \max_{n\in N}\left(EH_{min}/\gamma T\left(P_{n}h_{n,n}+I_{C}^{n}\right)\right) \end{array}$$
where $A=\sum_{n\in N}-\log_{2}\left(1+SINR_{n}^{D}\right)$, $B=\sum_{n\in N}\left(1-\tau_{n}^{ba}\right)\log_{2}\left(1+SINR_{n}^{D}\right)$, $C=\sum_{n\in N}$$\left(-p_{n}T-\gamma T\left(p_{n}h_{n,n}+I_{C}^{n}\right)\right)$, $D=\sum_{n\in N}2p_{cir}T+p_{n}\left(1-\tau_{n}^{ba}\right)T$. Next, we can equivalently solve the following linear programming using the method in [40]:(30)$$\begin{array}{cc} \; & \underset{\overset{´}{\eta },t}{max}A\overset{´}{\eta }+Bt\\ s.t.C1 & :\overset{´}{\eta }\leq \min_{n\in N}\left(1-\tau_{n}^{ba}-R_{D}^{min}/\log_{2}\left(1+SINR_{n}^{D}\right)\right)t\\ C5 & :\overset{´}{\eta }\leq \min_{n\in N}\left(1-\tau_{n}^{ba}\right)t\\ C7 & :\overset{´}{\eta }\geq \max_{n\in N}\left(EH_{min}/\gamma T\left(p_{n}h_{n,n}+I_{C}^{n}\right)\right)t\\ C8 & :C\overset{´}{\eta }+Dt=1\\ C9 & :\overset{´}{\eta }\geq 0,t\geq 0 \end{array}$$
where $\overset{´}{\eta }=\frac{\tau_{e}}{C\tau_{e}+D}$ and $t=\frac{1}{C\tau_{e}+D}$. Problem (Equation 30) is obviously convex over $\left(\overset{´}{\eta },t\right)$ and can be solved by the interior point method [39], and finally, the optimal value for $\tau_{e}$ is calculated as: $\tau_{e}=\overset{´}{\eta }/t$.

### 4.4. Alternating Optimization and Convergence Analysis

After the solution to each subproblem has been obtained, we give an overall algorithm for solving Problem (Equation 15). The original variables $\phi ,p,\tau_{e}$ can be divided into three blocks and alternately optimized. During each iteration, we iteratively optimized one of the variables by keeping the other two fixed, and the optimized variable would be delivered as the input of the next block. The detailed process for the alternating optimization is illustrated in Algorithm 3.
**Algorithm 3** Alternating optimization for solving Problem (Equation 30)1:**Initialize** iteration index l=0 and the $\left(\phi^{l},p^{l},\tau_{e}^{l}\right)$ satisfying the constraints in (Equation 15).2:**while** The improvement of η is higher than a predefined threshold $\epsilon_{2}$
**do**.3:    Solve Problem (Equation 16) given $\left(p^{l},\tau_{e}^{l}\right)$ and obtain the optimal solution $\phi^{l+1}$.4:    Solve Problem (Equation 27) given $\left(\phi^{l+1},\tau_{e}^{l}\right)$ and obtain the optimal solution $p^{l+1}$.5:    Solve Problem (Equation 30) given $\left(\phi^{l+1},p^{l+1}\right)$ and obtain the optimal solution $\tau_{e}^{l+1}$.6:    Update l=l+1.7:**return** The optimal beamwidth $\phi^{*}$, transmit power $p^{*}$, and EH time ratio $\tau_{e}^{*}$.

Next, we give the convergence analysis of the proposed algorithm. Let $\eta^{l}=\eta \left(\phi^{l},p^{l},\tau_{e}^{l}\right)$ denote the objective value after the *l*th iteration. First, in the (l+1)th iteration, there exist $\eta \left(\phi^{l},p^{l},\tau_{e}^{l}\right)\leq \eta \left(\phi^{l+1},p^{l},\tau_{e}^{l}\right)$ after performing beamwidth optimization. The reason lies in that $\phi^{l}$ is the input of Algorithm 1 fixing $\left(p^{l},\tau_{e}^{l}\right)$, and the switch rule of Algorithm 1 guarantees the non-decreasing property of the objective value. Second, we have $\eta \left(\phi^{l+1},p^{l},\tau_{e}^{l}\right)\leq \eta \left(\phi^{l+1},p^{l+1},\tau_{e}^{l}\right)$ due to the $p^{l}$ being the input local point, and the optimization of Problem (Equation 27) ensures that $p^{l+1}$ can achieve the non-decreasing objective value. Third, $\eta \left(\phi^{l+1},p^{l+1},\tau_{e}^{l}\right)\leq \eta \left(\phi^{l+1},p^{l+1},\tau_{e}^{l+1}\right)$ holds since Problem (Equation 30) is solved optimally. Finally, we can conclude that $\eta \left(\phi^{l},p^{l},\tau_{e}^{l}\right)\leq \eta \left(\phi^{l+1},p^{l+1},\tau_{e}^{l+1}\right)$, which means the objective value of problem (Equation 15) is non-decreasing after each iteration. Since the three variables of (Equation 15) are bounded by the constraints and the upper bound of the Problem (Equation 15) exists, we can conclude that the convergence of the proposed algorithm is guaranteed.

### 4.5. Complexity Analysis

First, we assumed the proposed coalition game in the beam alignment phase needs $N_{1}$ iterations to converge. During each iteration, 4N times of calculation are needed for DUs to calculate the throughput, harvested energy, power consumption, and beam alignment ratio, respectively, and *M* times of the calculation are needed for CUs to verify the minimum rate requirement. Hence, the complexity of beam alignment phase is $O(N_{1}\left(4N+M\right))$. In the power optimization phase, the complexity to solve Problem (Equation 27) using the interior point method [38] is $O(N^{3})$. Hence, the complexity for power optimization is $O(N_{2}N^{3})$, where $N_{2}$ is the number of iterations for Algorithm 2. Similarly, in the EH time ratio optimization phase, the complexity is O(1) using the interior point method, due to $\left(\overset{´}{\eta },t\right)$ being the only two variable to be optimized. Hence, the total complexity of our algorithm is mainly related to the beam alignment phase and the power optimization phase, which can be calculated as: $O(N_{OA}\left(N_{1}\left(4N+M\right)+N_{2}N^{3}\right))$, where $N_{OA}$ is the number of iteration for the overall algorithm. It should be pointed out that the practical running time of the algorithm depends on the settings of specific network parameters and the convergence threshold; hence, we further evaluate the time complexity in the next section.

## 5. Numerical Results

In this section, we present our numerical results to verify the convergence and effectiveness of the proposed algorithm. We considered a heterogeneous mmWave air-to-ground network scenario, where 10 UBSs were deployed above a circular area with a radius of 200 m [21]. The height of the UAVs was fixed at 100 m [41]. Multiple CUs and EH-enabled D2D pairs were randomly distributed within the target area. It is worthwhile to note that we assumed the power for each BS was fixed to 46 dBm, and we also set the maximum distance for each D2D link $D_{n}^{T}$−$D_{n}^{R}$ to be 50 m. The minimum beamwidth for DU-$\phi_{min}$ was set to $10^{\circ }$. The convergence criteria $\epsilon_{1}$ and $\epsilon_{2}$ were set to $10^{-3}$. The other simulation parameters are shown in Table 1.

To verify the effectiveness of our algorithm, we compared the performance of the proposed algorithm with the following algorithms:ES: exhaustive search, which traverses each possible beamwidth combination of DUs in the beamwidth selection phase. ES can obtain the optimal solution to beamwidth selection, but incurs an unbearable time complexity;PSO [31]: the particle swarm optimization algorithm, which forms multiple feasible beamwidth solutions to continuously evolve along the direction of increasing the utility until reaching the local optimal/global optimal solution. The parameter in the execution of PSO was the same as in [31];GA [42]: the genetic algorithm, which is also a population-based method. The GA evolves to the optimal/sub-optimal solution using the operations of mutation and crossover. In the simulation, the parameters including the population size and the probability for mutation and crossover were set the same as in [42];FBFP: the fixed beamwidth and fixed power strategy, in which the beamwidths and transmit power of all the D2D pairs are both fixed to a constant value and not varied with time;Reference [10]: Joint Optimization of the EH Time Ratio and Beamwidth (JOETRB). However, the interference between users was ignored, and the power optimization was not considered.

It is worth noting that ES, PSO, and the GA were only used in the beamwidth selection phase in our benchmark schemes. In the phase for optimizing p and $\tau_{e}$ of the two algorithms, our proposed power optimization and time scheduling algorithm were adopted. The convergence analysis of ES, PSO, and the GA combined with the proposed power optimization and time scheduling was similar to our proposed algorithm; hence, we omit it here. It should also be noted that all the following tables and curves are the average result of 100 independent experiments.

In Figure 3, fixing M=10 and the threshold of convergence $\epsilon_{2}$ to $10^{-3}$, we show the convergence behavior of the proposed algorithm. Under N=10, the proposed algorithm can converged within seven iterations. When the number of DU was increased to two, the number of iteration to converge was slightly increased to nine. This indicates that our proposed algorithm can converge within a small number of iterations and the fast convergence speed of the proposed algorithm is verified.

In Table 2, we list the time complexity of the posed CG, PSO, GA and ES, where $S_{PSO}$ and $S_{GA}$ denote the population size for PSO and Ga, $N_{PSO}$ and $N_{GA}$ denote the maximum number of iterations to converge for PSO and GA. It can be seen that the proposed CG greatly reduced the complexity compared with ES, which had exponential complexity. Generally speaking, the execution of PSO and the GA required a large population and number iterations to converge. Therefore, the complexity of the proposed CG also outperformed PSO and the GA.

In Figure 4, we study the MATLAB software running time of the beam alignment phase of the algorithms under different θ and *N*, to compare the time complexity of the proposed algorithm with ES and PSO. Obviously, the total time for the convergence of CG under different settings was much shorter than PSO and ES. For all the algorithms, when θ increased from $45^{\circ }$ to $60^{\circ }$, the required time to converge increased due to the expansion of the search range for the beamwidth. The figure also shows that when the number of DUs increased from 10 to 20, the time to converge for PSO under $\theta =60^{\circ }$ increased from 26.44 s to 82.54 s, the time to converge for the GA increased from 29.36 s to 87.94 s, while for CG, under $\theta =60^{\circ }$, the convergence time only increased from 8.29 s to 20.63 s. Among the three algorithms, ES showed the worst-case scenario. As every candidate combination of beamwidths for the DUs needed to be evaluated, the total running time for ES was unacceptably long. Moreover, our proposed algorithm could obtain better performance than PSO. As a great number of candidate solutions needed to be evaluated and updated, the total time for PSO and the GA to converge was far over CG. Hence, the superiority of the CG-based beamwidth selection algorithm with regard to time complexity was verified. The proposed algorithm can obtain the beamwidths for the DUs in a short time, thus effectively increasing the transmission time.

Next, we evaluated the EE performance comparing the proposed algorithm with the benchmark schemes. To find the parameters that may affect the network EE, we compared the network EE versus the number of DUs *N*, the number of CUs *M*, the maximum power for DU $P_{D}^{max}$, the minimum harvested power $EH_{min}$, and the minimum rate for CU $R_{min}^{C}$, respectively.

In Figure 5, we fix M=10 and plot the network EE versus the number of D2D links, comparing the proposed algorithm with the benchmark schemes under different *N*. It can be seen that the proposed algorithm can achieve better performance than FBFP and a performance near PSO and ES. The beamwidth selection and power control in our algorithm were designed to improve the network EE, and the EH time ratio optimization could further increase the EE. Another observation is that the network EE decreased with *N*. The reason lied in that although the narrow beams were adopted for the DUs, more D2D transmitters would potentially increase the interference between the DUs, thus degrading the transmission rate of the DUs, which degrades the network EE. A similar observation can be found by comparing the EE of the JOETRB [10] and FBFP algorithms under different *N*. However, it can be seen that the EE for the JOETRB [10] and FBFP sharply decreased when *N* exceeded 15, while our proposed algorithm still maintained a good level thanks to the dynamic adjustment of the beamwidths, power levels, and EH time ratio for the DUs.

In Figure 6, we plot the network EE versus different numbers of CUs to compare the performance of the proposed algorithm with ES and FBFP under different *M*. It can be seen from the figure that the energy efficiency of the proposed algorithm and the benchmark schemes decreased with the increase of *M*. The reason was that the growing of *M* led to the increased interference from the CU, i.e., $I_{C}^{n}$. At the same time, in order to meet the minimum rate constraint for the CUs, the DUs had to strictly control the transmit power, which degraded the transmission rate for the DUs. Although the growth in the number of CUs provided more energy for the DUs to harvest, it also led to the decrease of the user rate, which significantly impacted the EE. Therefore, in a heterogeneous network where CUs and DUs coexist and share the spectrum resources, the EE performance of the DUs will be degraded when the number of CUs increases. It can be seen from the figure that the performance of the proposed algorithm still approached PSO, the GA, and ES and outperformed the JOETRB [10] and FBFP strategies, which further verified the effectiveness of the proposed algorithm.

In Figure 7, we change the maximum transmission power of the DU and plot the curve of EE. With the increasing of $P_{D}^{max}$, some users in the network would increase their power to improve the user rate. Therefore, the EE first increased with $P_{D}^{max}$. However, due to the existence of a minimum rate for the CU $R_{min}^{C}$, the EE did not always increase, but was upper bounded by a certain value. When $P_{D}^{max}$ increased from 19 dBm to 23 dBm, the network EE increased. However, the EE began to fall when $P_{D}^{max}$ was further increased to 25 dBm. In addition, we also set the fixed power under the JOETRB [10] and FBFP to $P_{D}^{max}$ to find the relationship between the EE and $P_{D}^{max}$ under FBFP, and a similar finding could be obtained. Hence, we can reach the conclusion that the network EE was closely related to the transmit power of the DUs and first increased, then decreased with $P_{D}^{max}$.

In Figure 8, we evaluate the effect of $EH_{min}$ on the network EE. As $EH_{min}$ increased, the network EE gradually increased, which can be seen from the curves of the proposed algorithm, PSO, and ES. However, with the further growing of $EH_{min}$, the network EE no longer had unlimited growth. The reason was that the increased $EH_{min}$ required the DUs to harvest more energy, so the DUs were more inclined to increase the power and increase the EH time ratio $\tau_{e}$. However, the EE will not monotonically increase with the EH time ratio, because with more energy harvested, less energy will be consumed and the user rate will decrease at the same time due to the reduction of the effective transmission time. Moreover, due to the existence of the $R_{min}^{C}$, the power of the DUs was also limited. It can be seen that the proposed algorithm could achieve a performance close to PSO, the GA, and ES.

In Figure 9, we plot the network EE under different $R_{min}^{C}$. As $R_{min}^{C}$ increased, the EE of all three algorithms decreased. The reason was that when $R_{min}^{C}$ increased, the DUs had to decrease their transmission power to reduce the interference to the CUs, so as to meet the rate requirement of the CUs. By comparing the performance of the three algorithms, a similar conclusion can be drawn that the proposed algorithm can achieve a close performance to PSO, the GA, and ES.

## 6. Conclusions

EH-enabled D2D communication has shown great potential to be applied in the 5G network and beyond. In this paper, we investigated the energy efficiency of D2D users in a mmWave A2G network consisting of CUs and DUs served by UAV-BSs. We aimed to maximize the network energy efficiency, while guaranteeing the rate requirements of the CUs and DUs. The problem was formulated as the joint optimization of the beamwidth, transmit power, and EH time ratio of the D2D users. Alternating optimization was adopted to iteratively optimize one of the variables, fixing the other two. Firstly, a non-cooperative coalition game model was established to adjust the beamwidths of the DUs. Next, to tackle the non-convexity in the subproblem of power control, we exploited the methods of Dinkelbach and successive convex approximation. Finally, the EH time ratio optimization was performed by using linear fractional programming. The simulation results showed that our proposed algorithm could achieve better performance compared to the scheme with a fixed beamwidth and power and achieve a performance close to PSO, the GA and ES, but greatly reduce the time complexity. Moreover, the convergence of our proposed algorithm was also validated.

## Figures and Tables

**Figure 1 entropy-24-00300-f001:**
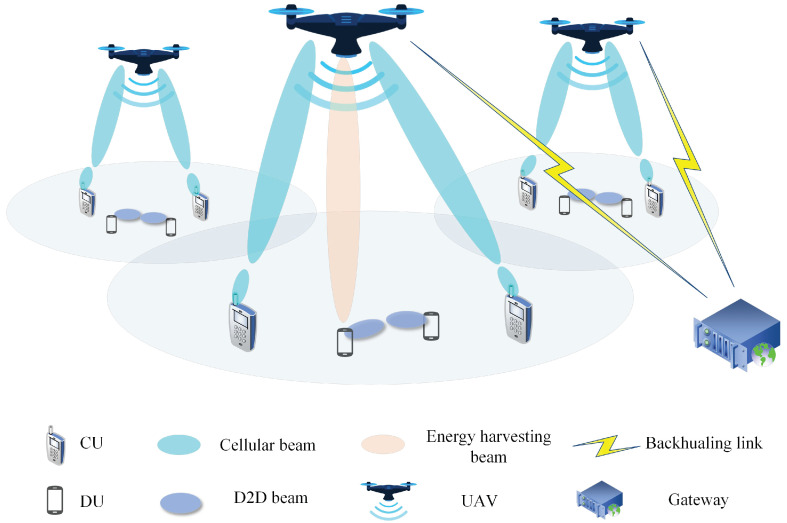
Network architecture.

**Figure 2 entropy-24-00300-f002:**
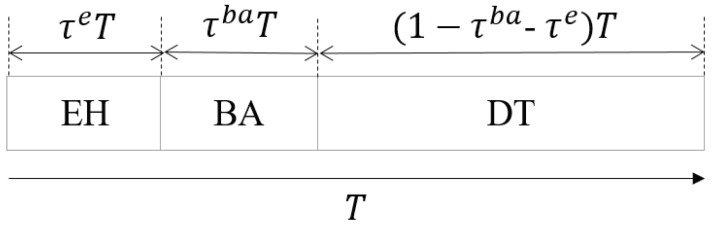
Time frame paradigm for the EH-BA-DT protocol, where the actual time ratio for EH, BA, and the DT is determined by the proposed algorithm.

**Figure 3 entropy-24-00300-f003:**
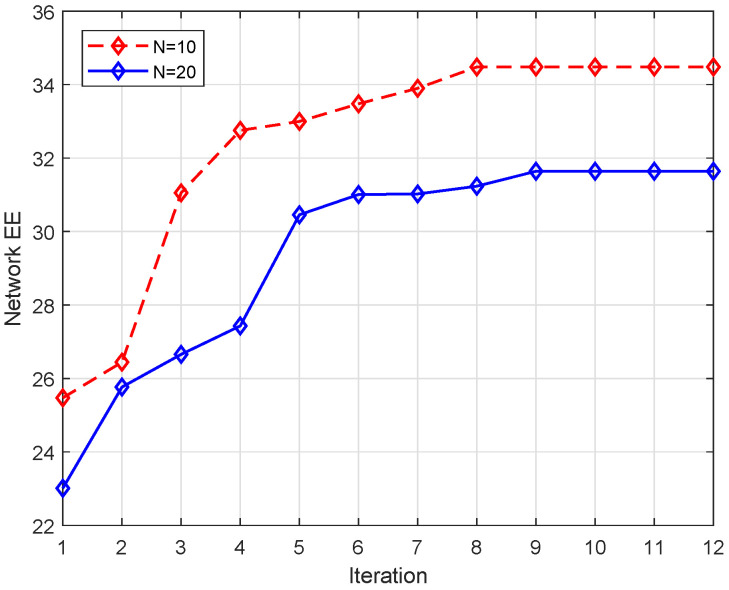
Convergence of the proposed algorithm.

**Figure 4 entropy-24-00300-f004:**
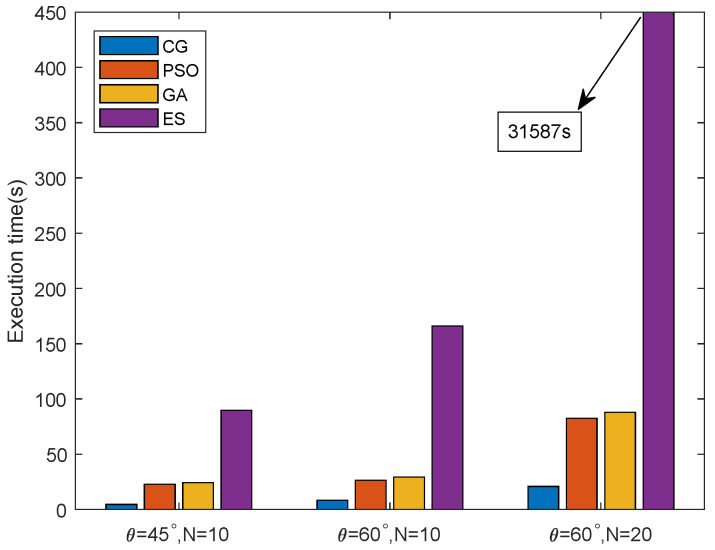
Running time for the algorithms in the beam alignment phase.

**Figure 5 entropy-24-00300-f005:**
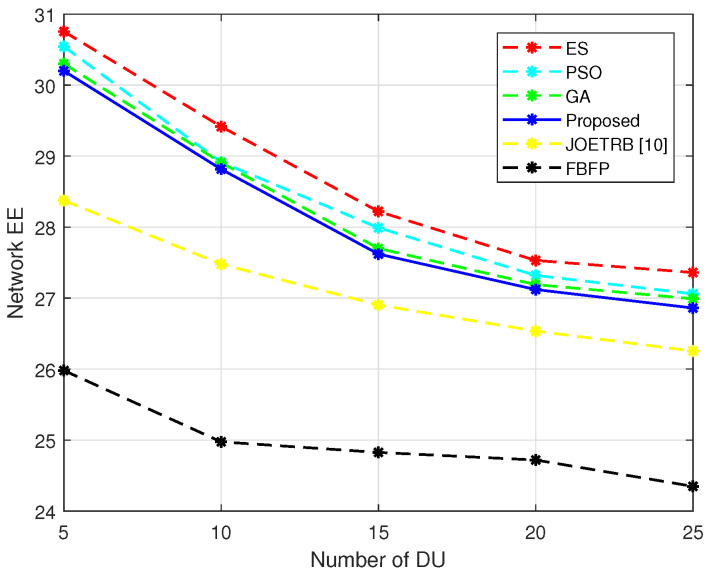
Network EE versus different *N*, M=10.

**Figure 6 entropy-24-00300-f006:**
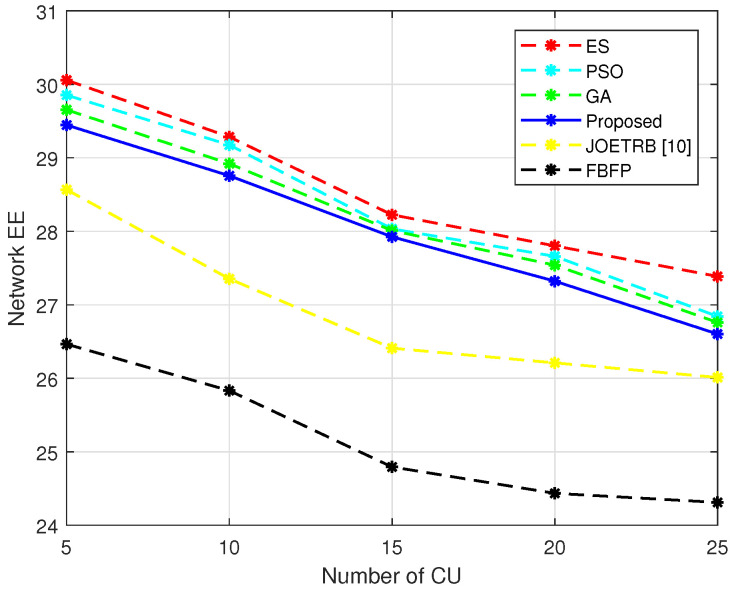
Network EE versus different *M*, N=10.

**Figure 7 entropy-24-00300-f007:**
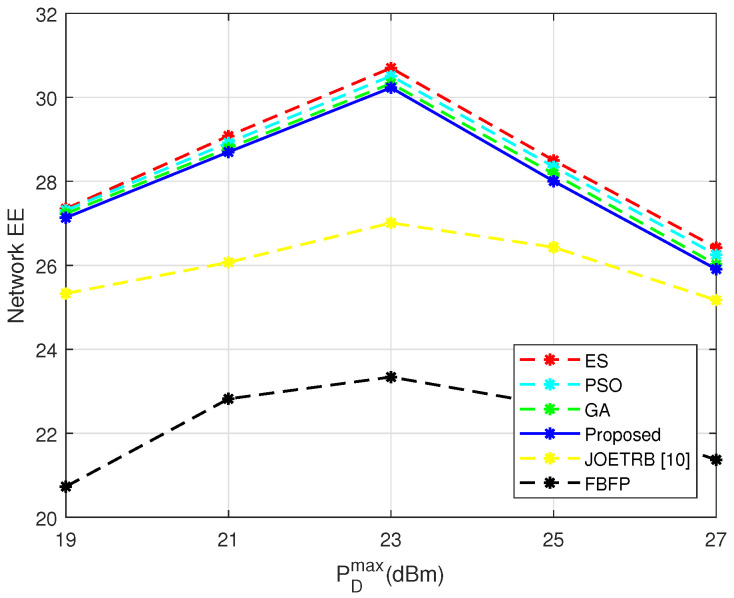
Network EE versus different $P_{D}^{max}$, M=10 and N=10.

**Figure 8 entropy-24-00300-f008:**
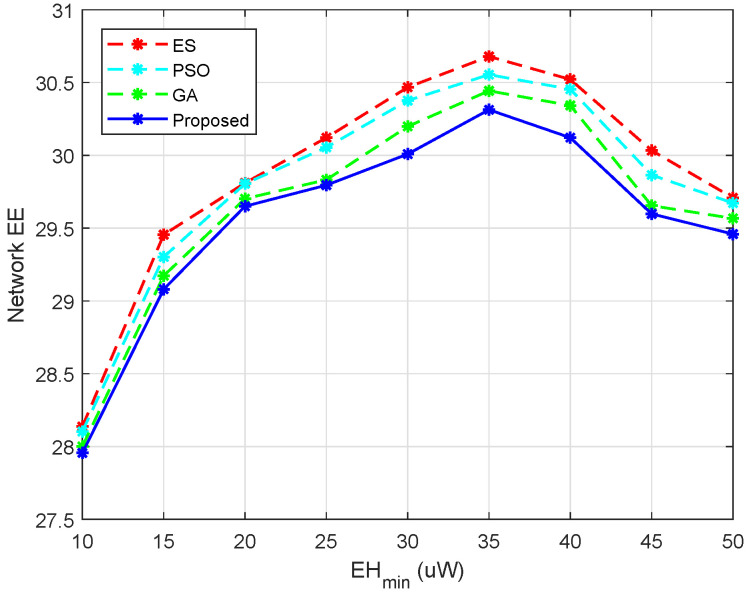
Network EE versus different $EH_{min}$, M=10 and N=10.

**Figure 9 entropy-24-00300-f009:**
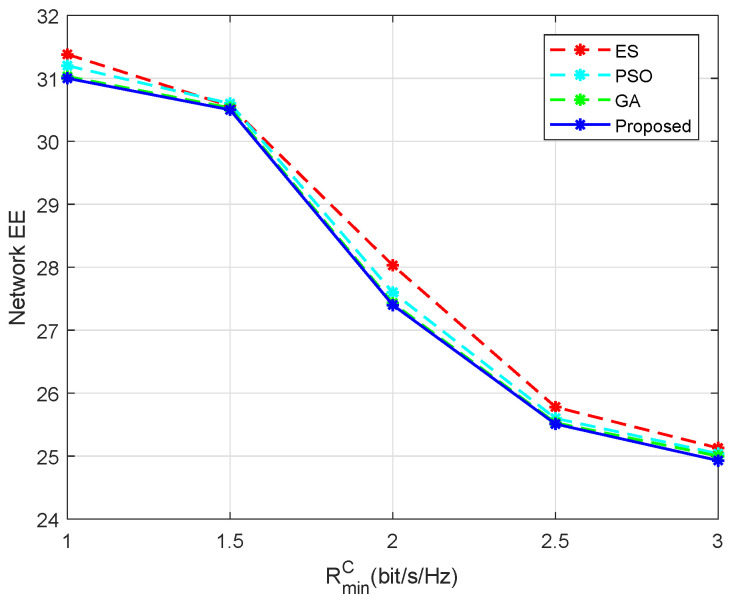
Network EE versus different $R_{min}^{C}$, M=10 and N=10.

**Table 1 entropy-24-00300-t001:** Simulation parameters.

Radius of target area [21]	200 m
Carrier frequency [10]	60 GHz
Pilot transmit/slot time [10], $T_{p}/T$	0.0001
Sidelobe antenna gain [36], *z*	0.05
Reference distance [36], $d_{0}$	1.5 m
Path loss exponent [36], *A*	2
Shadow fading variance [36], $\sigma_{SF}^{2}$	6 dB
Circuitry power [21], $P_{cir}$	20 dBm
Maximum transmit power for DU [21], $P_{D}^{max}$	23 dBm
Energy harvesting efficiency [23], γ,	0.5
Minimum rate for CU [21], $R_{min}^{C}$,	2 bit/s/Hz
Minimum rate for DU [21], $R_{min}^{D}$,	1 bit/s/Hz

**Table 2 entropy-24-00300-t002:** Comparison of the time complexity of different algorithms for beamwidth selection.

Proposed CG	$O(N_{CG}\left(4N+M\right))$
PSO	$O(N_{PSO}S_{PSO}\left(4N+M\right))$
GA	$O(N_{GA}S_{GA}\left(4N+M\right))$
ES	$O(N^{\left|\mathrm{BW}\right|}\left(4N+M\right))$

## Data Availability

Data, models, or code that support the findings of this study are available from the authors upon reasonable request.

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
