# Peer review of "Energy-Efficient Optimization for Energy-Harvesting-Enabled mmWave-UAV Heterogeneous Networks"

_entropy, 2022, doi:10.3390/e24020300_

Round 1
Reviewer 1 Report
The paper titled “Joint Beamwidth Selection and Power Control for EH-enabled D2D Underlying UAV-assisted mmWave Networks” uses joint optimization to solve for both power control problem and beamwidth allocation for energy harvesting (EH) equipped D2D communications, delivering a low complexity solution.
++ strong point:
The work is well written and easy to follow
The mathematical derivations are well presented and thorough
— weak points:
The discussions could be extended with adding more benchmarks for instance
Too many typos The system model seems a bit hypothetical.
Comments:
Which technology are we targeting and is this solution scalable to other technologies? In fact, it is not very clear why the simulations are done in 60 GHz?
In figure 2, EH gets a higher portion of time compared with BA. Is this intentional? If not, the caption could be extended to avoid confusion of the reader.
Theorem 1 is a too trivial to be counted as original contribution, it could be moved to appendix or stay in the main body as is, depending on what the authors prefer, however they can attempt to cut the length of the paper shorter to find a balance between the content of the work and the length of the presentation.
In the simulations, the EH efficiency factor is assumed to be 0.9. What is the reference for choosing this number? In the existing work, “Efficiency of DC Combination of Rectified Waveforms in Energy Harvesting Systems” the conversion efficiency does not go above 0.5.
Which implementation of PSO is used here? How does the implementation choice effect the performance, in particular the time complexity?
The benchmarking includes ES and PSO which are proper choices to give the reader an understanding of how the proposed method performs. Authors did not provide genetic algorithms (GAs) as another benchmark. They are recommend to choose an implementation of GAs and update the comparisons.
Knowing that reducing time complexity is the highlight of this work, section 3.5 is underdeveloped. Running times in Figure 4 are good illustrations but are not sufficient. A table could be added here, listing the O() notation for time complexity of PSO, ES, etc … so the reader could benefit from a mathematical description of how fast this method outperforms others.
On the writing side: **There should be a space between a work and a character, here are examples all over the paper: **harvesting(EH) ==> harvesting (EH) **array[8] ==> array [8] **OAM(Operation and Management) ==> OAM (Operation and Management) **coalition game(CG) ==> coalition game (CG) **Theorem 3[38] ==> Theorem 3 [38] **There is an extra parenthesis at the end of the keywords **The Gaussian distribution defined after equation 7 is missing an “N” **On page 13, l−th iteration and (l+1)−th iteration, the “th” should be moved to superscript
Author Response
Dear reviewer,
Feel your valuable guidance. I have modified and adjusted my manuscript based on your comments. The revised manuscript incorporates all the reviewers' comments. For each modification of the original manuscript, we have highlighted it in the revised version, and the line number or position description can be seen in the cover letter. The following is the reply to your comments:
Q:Which technology are we targeting and is this solution scalable to other technologies? In fact, it is not very clear why the simulations are done in 60 GHz?
A:In this paper, we want to solve the energy efficiency problem in mmWave UAV networks, and want to get improved energy efficiency at the cost of low time complexity. The proposed solution based on alternating optimization is also widely used in other optimizations in wireless networks. We choose 60GHz as the frequency in the simulation, the reason is that 60GHz is a widely used millimeter wave frequency, which is also consistent with the setting in the reference [26] [31][34].
Q:In figure 2, EH gets a higher portion of time compared with BA. Is this intentional? If not, the caption could be extended to avoid confusion of the reader.
A:In Figure 2, we show a schematic representation of the division of time frames, the time division in Figure 2 does not represent the actual time division, and we also explain this in the caption of Fig.2.
Q:Theorem 1 is a too trivial to be counted as original contribution, it could be moved to appendix or stay in the main body as is, depending on what the authors prefer, however they can attempt to cut the length of the paper shorter to find a balance between the content of the work and the length of the presentation.
A:We have changed the original Theorem 1 to be shown in the main text to avoid lengthy statement,please see page 9, line 258.
Q:In the simulations, the EH efficiency factor is assumed to be 0.9. What is the reference for choosing this number? In the existing work, “Efficiency of DC Combination of Rectified Waveforms in Energy Harvesting Systems” the conversion efficiency does not go above 0.5.
A:Before the major revision, we set a relatively ideal energy conversion efficiency (0.9) in the system model and simulation. However, after receiving the reviewers' comments, we referred to the energy harvesting efficiency settings in the literature, and finally determined the energy harvesting efficiency to 0.5, which is also the setting of reference [20]. We re-run simulation experiments based on the modified parameter.
Q:Which implementation of PSO is used here? How does the implementation choice effect the performance, in particular the time complexity?
A:We implement PSO based on the parameters of reference [32], and we also indicate it in the text, see page 14, line 331. In the implementation of PSO, the size of population and maximum iteration impact the time complexity. We compare the time complexity of proposed CG, PSO,GA and ES in Table 2, and illustrate the impact of the parameters of PSO on the time complexity, see page 14, line 352.
Q:The benchmarking includes ES and PSO which are proper choices to give the reader an understanding of how the proposed method performs. Authors did not provide genetic algorithms (GAs) as another benchmark. They are recommend to choose an implementation of GAs and update the comparisons.
A:Following the advice of the reviewer, we add the comparison algorithm of GA in the simulation and compare the performance of different algorithms in fig.4-fig.9.
Q:Knowing that reducing time complexity is the highlight of this work, section 3.5 is underdeveloped. Running times in Figure 4 are good illustrations but are not sufficient. A table could be added here, listing the O() notation for time complexity of PSO, ES, etc … so the reader could benefit from a mathematical description of how fast this method outperforms others.
A:Following the advice of the reviewer, we add table.2 to show the time complexity of all the algorithms and illustrate why the proposed CG outperforms the other algorithms in terms of time complexity.
Q:On the writing side: **There should be a space between a work and a character, here are examples all over the paper: **harvesting(EH) ==> harvesting (EH) **array[8] ==> array [8] **OAM(Operation and Management) ==> OAM (Operation and Management) **coalition game(CG) ==> coalition game (CG) **Theorem 3[38] ==> Theorem 3 [38] **There is an extra parenthesis at the end of the keywords **The Gaussian distribution defined after equation 7 is missing an “N” **On page 13, l−th iteration and (l+1)−th iteration, the “th” should be moved to superscript
A:We carefully checked the formatting irregularities in the paper and get them revised in accordance with the reviewers' comments.
Reviewer 2 Report
The paper presents an interesting approach to beamwidth selection and power control for EH-enabled UAV.
The methods, results and conclusion of this manuscript are well discussed.
The title is misleading please retitle the work in a more comprehensive manner.
Also recheck the paper for typos.
The simulation code must be shared with the community.
Author Response
Dear reviewer,
Feel your valuable guidance. I have modified and adjusted my manuscript based on your comments. The revised manuscript incorporates all the reviewers' comments. For each modification of the original manuscript, we have highlighted it in the revised version, and the line number or position description can be seen in the cover letter. The following is the reply to your comments:
Q: The title is misleading please retitle the work in a more comprehensive manner.
A: Following the advice of the reviewer, we have changed the title of the article to “Energy Efficient optimization for Energy Harvesting-enabled mmWave-UAV heterogeneous Networks”.
Q: Also recheck the paper for typos.
A: We carefully checked the typos and formatting irregularities in the paper and get them revised.
Q: The simulation code must be shared with the community.
A: Following the advice of the reviewer, the simulation code will be shared with the community.
Reviewer 3 Report
In this paper , the authors proposed ways of achieving energy efficiency of Device-to-Device (D2D) users in a mmWave air-to-ground (A2G) network consisting of cellular, and D2D users served by UAV-BSs. The goals is to archive the maximization of network energy efficiency. The authors have provided strong background studies, and details of related work. However, there is scope for improvement.
1. The topic is written with extensive proof, and equations, but it is difficult for the reader to understand the interconnection between them. For example, the three algorithms presented also talks on different parameters, such as bandwidth, power optimization, and alternating optimization. So, there must be clarity on describing the relationships, and how the time complexities of these algorithms (i.e., Coalition game-based beamwidth selection algorithm for D2D pairs during each iteration, SCA-based power optimization for D2D links during each iteration, and Alternating optimization for solving problem (30)) have impact on proposed maximization of network energy efficiency.
2. In the abstract the authors mentioned that, the “proposed algorithm outperforms the fixed beamwidth and fixed power strategy and can closely approach the performance of exhaustive search and Particle Swarm Optimization but with much reduced time complexity”. Even though authors mentioned PSO, and exhaustive search, there less description on ways of proving this with experimental results.
3. The English and writing of this paper is good. However, there are some acronyms need to be expanded during their first use such as : LoS (page 2), SINR (page 4) etc.
4.The font type of BW (feasible beamwidth set) is different in text, and in equation 2.
Author Response
Dear reviewer,
Feel your valuable guidance. I have modified and adjusted my manuscript based on your comments. The revised manuscript incorporates all the reviewers' comments. For each modification of the original manuscript, we have highlighted it in the revised version, and the line number or position description can be seen in the cover letter. The following is the reply to your comments:
Q: The topic is written with extensive proof, and equations, but it is difficult for the reader to understand the interconnection between them. For example, the three algorithms presented also talks on different parameters, such as bandwidth, power optimization, and alternating optimization. So, there must be clarity on describing the relationships, and how the time complexities of these algorithms (i.e., Coalition game-based beamwidth selection algorithm for D2D pairs during each iteration, SCA-based power optimization for D2D links during each iteration, and Alternating optimization for solving problem (30)) have impact on proposed maximization of network energy efficiency.
A: Thanks for reviewer’s advice, in page.2, line 63-line 76, we have clarified the relationship between optimizing the three variables(beamwidth, power and time ratio) and network EE.
Q: In the abstract the authors mentioned that, the “proposed algorithm outperforms the fixed beamwidth and fixed power strategy and can closely approach the performance of exhaustive search and Particle Swarm Optimization but with much reduced time complexity”. Even though authors mentioned PSO, and exhaustive search, there less description on ways of proving this with experimental results.
A: In fig.4, we have compared the running time for all the algorithms. Following the advice of the reviewer, to further validate the superiority of the proposed CG in terms of time complexity, we add table.2 to show the time complexity of all the algorithms in a mathematical description manner.
Q: The English and writing of this paper is good. However, there are some acronyms need to be expanded during their first use such as : LoS (page 2), SINR (page 4) etc.
A:We have carefully checked the typos, acronyms and other format irregularities in the paper and get typos revised and acronyms explained.
Q: The font type of BW (feasible beamwidth set) is different in text, and in equation 2.
A: We have unified the font type of all the BW, see page.6, line.190.
Reviewer 4 Report
This paper considers a scenario where multiple unmanned aerial vehicles are deployed to provide energy for D2D users (DUs) and data transmission for cellular users (CUs). The proposed topic is very hot, and the paper shows interesting results. Furthermore, the paper is well organized and structured. However, the authors should improve the technical quality of the paper by following these recommendations:
-The proofreading is required. i.e. In the abstract paradiam should be “paradigm”.
-“Device-to-Device (D2D) communication integrating with EH can overcome battery-capacity shortage and improve energy efficiency (EE) by performing EH from ambient RF signals” should be reformulated.
-How the used UAVs can be helpful in assisting the ground users in energy harvesting knowing that the UAVs are energy-constrained too.
-What are the limitations of the coverage distance, and hovering?
-How can the realized simulations approach reality, regarding the mobility of the CU/DU and the drones?
-In other words, is the effect of mobility taken into consideration in the study?
-How might this impact the results?
-The choose of the simulation parameters in Table 1 should be explained.
-The proposed scenarios should be compared to others in the literature.
-The obtained results should be compared directly to other of the recent state of the art.
Author Response
Dear reviewer,
Feel your valuable guidance. I have modified and adjusted my manuscript based on your comments. The revised manuscript incorporates all the reviewers' comments. For each modification of the original manuscript, we have highlighted it in the revised version, and the line number or position description can be seen in the cover letter. The following is the reply to your comments:
Q:The proofreading is required. i.e. In the abstract paradiam should be “paradigm”.
A: We have carefully checked the paper and revised the typos, e.g. “paradiam”->”paradigm”.
Q:“Device-to-Device (D2D) communication integrating with EH can overcome battery-capacity shortage and improve energy efficiency (EE) by performing EH from ambient RF signals” should be reformulated.
A: We have reformulated this sentence to “Device-to-Device (D2D) communication can assist devices overcoming the disadvantage of limited battery capacity and improving energy efficiency (EE) by performing EH from ambient wireless signals.”
Q:How the used UAVs can be helpful in assisting the ground users in energy harvesting knowing that the UAVs are energy-constrained too.
A: In this paper, we don’t focus on the energy consumption of UAVs but the energy consumption of devices. Therefore, we assume that the D2D device are equipped with energy harvest technology and can perform EH from the wireless signal transmitted by UAVs . Hence, in this paper, we don’t focus on the energy constraint of UAVs since they have battery capacity than mobile devices and can be charged by charging station. However, this will be our future research direction. We explained this issue in the paper, see page.5, line.178
Q:What are the limitations of the coverage distance, and hovering?
A: In this paper, we follow the same assumption of [40] and assume that the coverage of the UAV is determined based on the pathloss threshold. The association between UAVs and users are established based on minimum pathloss criterion. In addition, we assume all the UAVs hover at the fixed altitude of 100m, following the same setting of [41]. We clarified this issue in the revised paper, see page. 5, Line.172.
Q:How can the realized simulations approach reality, regarding the mobility of the CU/DU and the drones? In other words, is the effect of mobility taken into consideration in the study? How might this impact the results?
A: In our paper, we consider a static scenario. However, our algorithm can also be utilized in the dynamic and fast-changing environment since the time frame can be approximately discredited into multiple snapshots, in which our system model and algorithm can be applied. We clarified this issue in page.5, line.174.
Q:The choose of the simulation parameters in Table 1 should be explained.
A: Following the advice of the reviewer, we have listed the reference for choosing the parameter in table.1.
Q:The proposed scenarios should be compared to others in the literature.
A: In our paper, we consider a mmwave-enabled D2D scenario, which is also considered in [25][26][30][31]. We also consider the combination of UAV, mmWave and EH, which is also studied in [12]-[16]. However, the combination of UAV, D2D, EH and mmwave is rarely studied. Therefore, the proposed scenario is different from the others in literature, as stated in page.3 line.88.
Q:The obtained results should be compared directly to other of the recent state of the art.
A: Following the advice of the reviewer, we add the genetic algorithm and reference [31] as benchmark schemes and compare the performance of all the algorithms in the simulation section.
Round 2
Reviewer 1 Report
Remove reference [31] from figure legends and replace it with a short scientific term please.
Also please go over the paper for punctuation errors and writing typos.
Other than this, all my concerns are addressed. Thank you.
Author Response
Dear reviewer:
Feel your valuable guidance. We have revised the manuscript according to your comments. The changes to the manuscript include:
- We rename the abbreviation of the algorithm in [31] as “JOETRB” (Joint optimization of EH time ratio and beamwidth), and the legends in the fig.5-fig.7 are also changed to “JOETRB”.
- We have checked and rectified all the typos and wrongly used punctuations in the formula, algorithm and main text.
Reviewer 4 Report
The authors have satisfactorily addressed my concerns.
The authors have done several efforts to improve both the presentation and the technical qualities of the paper.
Author Response
Dear editor and reviewers:
Feel your valuable guidance. We have revised the manuscript according to your comments. The changes to the manuscript include:
- We have checked and rectified all the typos and wrongly used punctuations in the formula, algorithm and main text.